# Epigenetic and Epitranscriptomic Antiviral Responses in Plants for Disease Management

**DOI:** 10.3390/v18010017

**Published:** 2025-12-22

**Authors:** Islam Hamim, Sadman Jawad Sakib, Md. Readoy Hossain, Jaima Noor Hia, Maria Hasan, Alvi Al Muhimine, John S. Hu

**Affiliations:** 1Department of Plant Pathology, Bangladesh Agricultural University, Mymensingh 2202, Bangladesh; 2Department of Plant and Environmental Protection Sciences, University of Hawaii at Manoa, Honolulu, HI 96822, USA

**Keywords:** epigenetic gene silencing, epitranscriptomic regulation, RNA-directed DNA methylation (RdDM), post-transcriptional gene silencing (PTGS), small interfering RNAs (siRNAs), plant viral disease, host–virus interaction

## Abstract

Plant viral diseases cause significant agricultural losses worldwide and are shaped by complex virus-host and virus-virus interactions. Unlike fungal or bacterial pathogens, viruses cannot be directly controlled with chemicals, and their management relies on insect vector control and the development of virus-resistant plant varieties. Plants deploy endogenous epigenetic (DNA/chromatin-based) and epitranscriptomic (RNA-based) mechanisms to limit viral infections. RNA silencing pathways, particularly post-transcriptional gene silencing (PTGS) mediated by small RNAs, restrict viral replication and shape viral populations. Additional layers, including RNA-directed DNA methylation (RdDM), N6-methyladenosine (m^6^A) RNA modifications, histone modifications and chromatin remodeling, further modulate host–virus interactions. DNA methylation can be inherited and may confer resistance to future generations, although its stability is partial and context-dependent. Virus-derived 24-nt small interfering RNAs (siRNAs) act as mobile signals, inducing systemic gene silencing and potentially influencing viral population dynamics. Understanding these epigenetic and epitranscriptomic mechanisms can improve virus diagnosis, pathogenesis studies and disease management, while also providing insights into viral diversity and co-infection dynamics. This review synthesizes current knowledge of these mechanisms and discusses their implications for developing sustainable antiviral strategies.

## 1. Introduction

Plant pathogenic viruses are among the most destructive agents threatening global crop production. Every year, they cause crop losses valued at approximately $30 billion [1], posing a serious threat to agricultural sustainability. The economic damage caused by plant viruses is well documented worldwide (Appendix A). Effective virus control is therefore critical for reducing losses and improving crop yields. Conventional methods such as insect vector control, have achieved limited success and being mostly chemical-based, may leave harmful residues in the environment. To address the evolving nature of viruses and develop more sustainable defense strategies, molecular approaches are increasingly being prioritized.

Plants respond to diverse biotic and abiotic stresses through adaptive mechanisms that involve epigenetic modifications (DNA methylation, histone modifications, chromatin remodeling) and epitranscriptomic RNA modifications, which regulate cellular and molecular functions [2]. These modifications can directly influence transcription and stress responsive gene expression, and in some cases, persist through cell divisions or are transmitted across generations [3,4].

Plant viruses can induce heritable epigenetic changes in host plants, influencing long-term defense responses. Viruses reprogram host cellular machinery by targeting proteins and pathways involved in defense, development, and signaling, thereby altering gene expression, disrupting the cell cycle, and impairing small RNA-mediated defenses. They further modulate host defense by interfering with DNA methylation and microRNA biogenesis pathways [5,6]. Viral infection can induce de novo or altered DNA methylation patterns [6,7,8], providing a relatively stable mechanism to regulate gene expression, including resistance-related genes [9], thereby influencing host susceptibility [10,11]. In addition, viruses alter histone modifications, which reshape chromatin structure and reprogram host gene expression [12].

Virus-derived small RNAs (vsRNAs) activate epigenetic and epitranscriptomic silencing pathways that suppress viral replication and gene expression [13]. Although sRNA-mediated silencing is a central component of plant antiviral immunity, similar small RNA–based mechanisms also contribute to defense against fungal and other pathogens, including through cross-kingdom RNA interactions [14,15,16,17]. Beyond antiviral and antifungal defense, sRNAs also regulate plant development, highlighting their dual role in gene regulation [16,17,18,19].

Despite significant progress in understanding plant epigenetic mechanisms, the role of epigenetic and epitranscriptomic gene silencing in defense against pathogenic viruses and its potential application in disease management remain poorly understood. Bridging this gap could enable innovative strategies to enhance crop resistance and reduce virus-related yield losses. This review summarizes advances in plant epigenetic and epitranscriptomic responses to viral infection, examines virus–host interactions that modulate these pathways, and explores their potential for improving crop protection. Such insights are vital for combating the virus-induced crop losses worldwide.

## 2. Epigenetic and Epitranscriptomic Antiviral Responses in Plants

Epigenetic gene silencing is a nucleus-centric host defense mechanism in plants that suppresses gene expression to maintain genome integrity. It protects host genomes from DNA viral infections and transposable elements [20,21]. Epigenetic silencing operates primarily through transcriptional gene silencing (TGS), which suppresses transcription via DNA methylation and histone modifications. In contrast, post-transcriptional gene silencing (PTGS) is an epitranscriptomic mechanism that acts largely in the cytoplasm to mediate the degradation of target RNA molecules, including RNA viruses [21,22].

### 2.1. Post Transcriptional Gene Silencing

Post-transcriptional gene silencing (PTGS) is an epitranscriptomic, sequence-specific manner defense mechanism in plants that targets viral RNAs in the cytoplasm, preventing the accumulation of viral proteins (Figure 1; Table 1) [12]. Unlike nucleus-centric epigenetic regulation, PTGS operates primarily at the RNA level and is especially critical for defense against RNA viruses.

Upon infecting a host, plant viruses initiate PTGS through the formation of double-stranded RNA (dsRNA). In RNA viruses, these dsRNAs are generated during viral genome replication, whereas in DNA viruses, dsRNAs often originate from bidirectional transcription of the viral genome [23]. Upon recognition of aberrant viral or host RNAs, RNA-dependent RNA polymerase 6 (RDR6) synthesizes dsRNA from these templates, acting downstream of initial RNA detection. RNA-dependent RNA polymerase 6 (RDR6) synthesizes dsRNA using viral single-stranded RNA (ssRNA) as a template. The resulting dsRNAs are subsequently processed into 21- and 22-nucleotide siRNAs by DICER-LIKE proteins DCL4 and DCL2, respectively [18,23]. Unlike DCL2/DCL4, DCL1 functions primarily in miRNA biogenesis and endogenous gene regulation, highlighting the mechanistic distinction between antiviral siRNA pathways and miRNA pathways [18,23,24].

These siRNAs primarily participate in PTGS; however, they can also feed into the RNA-directed DNA methylation (RdDM) pathway, inducing low levels of de novo DNA methylation via AGO2 and reinforcing transcriptional gene silencing (TGS), as illustrated schematically in Figure 1 ( Table 1). Following processing, siRNAs are incorporated into the RNA-Induced Silencing Complex (RISC) [7,23,24,25]. The double stranded siRNA unwinds upon association with RISC and usually only the guide (antisense) strand remains, forming the activated RISC. The activated RISC binds to complementary target mRNA, guiding siRNA to the target. ARGONAUTE 1(AGO1) then cleaves the mRNA at the binding site, executing PTGS [23,26,27]. Recent studies indicate that PTGS is tightly linked to mRNA decay pathways, reflecting a coordinated cytoplasmic antiviral surveillance system [28].

**Table 1 viruses-18-00017-t001:** Key components of PTGS and RdDM pathways in plants and their roles in antiviral defense.

Molecular Component	Category/Type	Associated Pathway (PTGS/RdDM)	Primary Function	Key References
DCL2, DCL4	Dicer-like RNase III enzymes	PTGS (DCL4/DCL2); RdDM (DCL2)	DCL4 makes 21-nt siRNAs for PTGS; DCL2 makes 22-nt siRNAs for antiviral silencing and non-canonical RdDM	[29,30,31]
DCL3	Dicer-like RNase III enzyme	Canonical RdDM	Processes dsRNA into 24-nt siRNAs that guide DNA methylation in the RdDM pathway	[32,33]
RDR6	RNA dependent RNA polymerase	PTGS, Non-canonical RdDM	Converts single-stranded RNAs to double stranded RNAs for processing into21–22 nt siRNAs by DCL1, DCL2 and DCL4	[34]
RDR2	RNA dependent RNA polymerase	Canonical RdDM	Converts Pol IV–derived ssRNAs into dsRNAs, which are processed by *DCL3* into 24-nt siRNAs.	[34,35]
AGO1	Argonaute protein	PTGS	Forms RISC with 21–22 nt siRNAs to recognize and cleave complementary mRNAs.	[34,36,37]
AGO2	Argonaute protein	Non-canonical RdDM	Loads 21–22 nt siRNAs to target Pol V transcripts and facilitates DRM2 recruitment for DNA methylation.	[29]
AGO4	Argonaute protein	Canonical RdDM	Loads 24-nt siRNAs to target Pol V transcripts and recruits DRM2 for DNA methylation.	[29]
NERD	Plant-specific protein (PHD and zinc-finger domains)	Non-canonical RdDM	Interacts with histone H3 and AGO2–siRNA complexes to promote histone modification and transcriptional repression.	[32]
DRM 2	DNA methyltransferase	Canonical & non-canonical RdDM	Catalyzes de novo cytosine DNA methylation guided by AGO–siRNA complexes.	[34]
HEN1	RNA methyltransferase	PTGS, All RdDM	Adds a 2′-O-methyl group to the 3′ end of siRNAs, protecting them from degradation	[38]

Remarkably, once established, PTGS can persist for extended periods, reflecting a form of epigenetic memory, although evidence for maintenance after complete viral clearance remains limited [39,40]. A well-known example of PTGS is the transgenic papaya resistance system against papaya ringspot virus (PRSV). This system employs a non-translated or inefficiently translated coat protein (CP) transgene, providing sequence-homology dependent resistance to the homologous PRSV strains [41,42].

### 2.2. Transcriptional Gene Silencing

Transcriptional Gene Silencing (TGS) occurs via RNA-directed DNA methylation (RdDM) and associated histone modifications, targeting both plant and viral genomes (Figure 1, Table 2). This epigenetic regulation takes place in the nucleus [19,43]. RdDM is the major small RNA-mediated epigenetic pathway in plants. This can occur either by a canonical or a non-canonical pathway involving small RNA biogenesis [32] (Figure 1). Our understanding of canonical RdDM pathway mainly originates from studies on the model plant *Arabidopsis thaliana* [34,44].

RNA Polymerase IV (Pol IV) directs the siRNA biogenesis by transcribing ssRNA at the target loci. Thus, Pol IV functions specifically in siRNA production, whereas Pol V functions in guiding the AGO–siRNA complex to target loci, clearly distinguishing their separate roles in canonical RdDM. RNA-DEPENDENT RNA POLYMERASE 2 (RDR2) then converts the ssRNA into dsRNA (Figure 1). The dsRNA is then processed by DICER-LIKE 3 (DCL3) into 24-nucleotide siRNAs, which are later stabilized by HUA ENHANCER 1(HEN1) [31]. The Pol IV-RDR2-DCL3 pathway is sufficient for the production of 24-nucleotide siRNAs in vitro, considering contributions from other associated factors [45].

For de novo methylation of DNA, a single strand from each 24-nucleotide siRNA is loaded into ARGONAUTE proteins AGO4, AGO6, or AGO9, forming the AGO–siRNA duplex. These AGO–siRNA complexes specifically bind to Pol V–derived scaffold transcripts, ensuring precise targeting. Subsequently, AGO4 recruits DOMAINS REARRANGED METHYLTRANSFERASE 2 (DRM2), which methylates the nearby DNA and suppresses the target gene expression [32,46,47]. Histone modifications including de-acetylation, de-methylation, and de-ubiquitination, further remodel chromatin at certain RdDM targets, achieving stable transcriptional gene silencing [31]. In plants, these are often associated with antiviral silencing [48,49], and arise primarily as byproducts of PTGS (Figure 1). The main function of non-canonical pathway is to establish silencing by sparsely methylating a targeted region, which later gets taken over by the canonical pathway to establish long term silencing [44]. Thus, the non-canonical pathway acts as a temporary bridge between PTGS and canonical RdDM pathway (Figure 2), targeting fewer genomic regions compared to the canonical pathway [29].

**Table 2 viruses-18-00017-t002:** Geminiviral protein-mediated suppression of host defense mechanism.

Viral Protein	Virus	Host Target	Effect on Epigenetic Gene Silencing	References
Rep (Replication-associated protein)	Tomato yellow leaf curl Sardinia virus (TYLCSV)	MET1, CMT3	Reduces maintenance DNA methylation (CG context), weakening transcriptional gene silencing (TGS)	[50]
AC2	Tomato golden mosaic virus (TGMV), Cabbage leaf curl virus (CaLCuV)	SUVH4/KYP (H3K9 histone methyltransferase)	Inhibits histone methylation, disrupting chromatin-based TGS	[51]
C2	Beet severe curly top virus (BSCTV)	SAMDC1 (S-adenosyl methionine decarboxylase)	Lowers methyl donor availability, reducing DNA and histone methylation for epigenetic silencing	[52]
C4	Tomato leaf curl Yunnan virus (TLCYnV)	DRM2 (Domain Rearranged Methyltransferase 2)	Prevents de novo cytosine methylation on viral DNA, impairing RdDM-mediated TGS	[53]
TrAP	TGMV, BCTV	ADK (Adenosine Kinase)	Disrupts SAM biosynthesis, interfering with methylation-mediated TGS	[54]
V2	TYLCV, Cotton leaf curl Multan virus (CLCuMuV)	AGO4	Blocks AGO4 binding to viral DNA, inhibiting RdDM and preventing transcriptional silencing	[24]
Pre-coat Protein	TYLCV, ToLCNDV	MET1, RDR1, HDA6	Suppresses maintenance methylation and chromatin silencing, compromising TGS	[55,56]
C4	CLCuMuV, ToYLCGDV	SAM synthetase, BAM1	Reduces SAM availability and inhibits TGS; disrupts epigenetic regulation of defense genes	[57,58,59]
AC5	MYMIV	CHH cytosine methyltransferase	Suppresses RNA-induced PTGS and reverses TGS of silenced transgenes, impairing epigenetic silencing	[60]
βC1	Betasatellite of TYLCCNV	SAHH (S-adenosyl homocysteine hydrolase)	Disrupts methyl cycle, suppresses methylation-dependent PTGS and RdDM-mediated TGS via calmodulin-like protein (CaM)	[61,62]
2b	Cucumber Mosaic virus (CMV)	AGO proteins	Disrupts siRNA mediated RdDM pathway and results in hypomethylation	[63]
HC-pro (Helper component protease)	Tobacco Vein Banding Mosaic Virus (TVBMV)	DNA methylation machinery	Decreases DNA methylation of the promoter of auxin biosynthesis genes targeted by RdDM and interferes with auxin signaling pathways	[64]
P19	Tombusvirus	AGO1	Enhanced accumulation of AGO1 and viral RNAs, thereby effectively suppressing the RNA silencing based host defense	[65]
F-box protein P0	Turnip Yellows Virus (TuYV)	AGO1	Blocks small RNA-mediated silencing and targets AGO1 via autophagy degradation processes, thus impairing PTGS mechanism of host	[66]

## 3. Plant Epigenetic (DNA-Based) and Epitranscriptomic (RNA-Based) Modifications

### 3.1. Plant DNA

DNA methylation occurs mainly at cytosine residues, which is the key epigenetic silencing system for regulating gene expression and maintaining genome stability in plants [67]. Methyl groups from S-adenosyl-L-methionine (SAM) are added to the 5th carbon of cytosine, generating 5-methylcytosine (5meC) on target sequences, including host gene promoters [43]. This modification is central to TGS in plants as defense against various pathogens, including viruses [68]. Mechanistically, it blocks transcription of adjacent genes, establishing TGS and restricting expression of viral or host genes [49]. In the canonical RdDM pathway dsRNA is processed by DCL3 into 24-nt siRNAs [69]. These siRNAs bind with AGO4/AGO6 guiding methyltransferases to target loci to induce de novo methylation [69]. In non-canonical pathway, DCL4 produces 21–22 nt siRNA which can bind with AGO4 and lead to DNA methylation [70]. DNA methylation occurs in symmetrical CG and CHG contexts and in asymmetric CHH contexts (where H = A, T, or C) [70,71].

Figure 2 schematically illustrates the context-specific DNA methylation patterns mediated by MET1, CMT3, CMT2, and DRM2. MET1 maintains symmetric CG methylation, while CMT3 catalyzes CHG methylation. CHH methylation is established by CMT2 and DRM2, with DRM2 responsible for de novo methylation. Together, these methyltransferases reinforce transcriptional gene silencing and contribute to plant antiviral defense. Within the canonical RdDM pathway, AGO4 binds virus-derived siRNAs and recruits DRM2 to viral DNA, promoting cytosine methylation that restricts geminivirus replication in Arabidopsis and other crop species [51].

### 3.2. Plant RNA

Adenosine methylation at the N6 position (m^6^A) is the most prevalent internal RNA modification in plants, occurring in rRNAs, mRNAs, tRNAs, miRNAs, and long noncoding RNAs [72]. About 80% of RNA methylation in plant cells is caused by m^6^A modifications [73]. Analyses show that m^6^A was found in wheat (*Triticum turgidum* L.), oat (*Avena sativa* L.) coleoptiles, and maize (*Zea mays* L.) [74,75,76]. However, the molecular mechanisms regulating m^6^A methylation in plants remain incompletely understood [77,78]. m^6^A modification is dynamically regulated by methyltransferases (writers), demethylases (erasers), and recognized by m^6^A binding proteins (readers) [77,78].

In plants, the core m^6^A writer complex consists of MTA and MTB as catalytic subunits, together with accessory proteins FIP37, VIRILIZER (VIR), and HAKAI, which are essential for successful methylation activity [79,80]. This modification plays a bidirectional role, it can inhibit viral RNA replication as part of the host antiviral defense, but in some cases, reversible m^6^A methylation may facilitate viral infection [79]. For example, virus infection can alter host mRNA methylation, affecting expression of defense-related genes and modulating stress responses [81]. When viral RNA enters the host plant, m^6^A methyltransferases such as plant METTL homologs catalyze methylation as part of a stress response [77,78,81]. Conversely, in watermelon infected with cucumber green mottle mosaic virus, ALKBH9B demethylases are activated, reducing m^6^A and promoting viral invasion (Figure 3).

Figure 3 illustrates how viral RNAs are dynamically regulated by m^6^A: methylated by ‘writers,’ demethylated by ALKBH9B ‘erasers,’ and recognized by YTH-domain ‘readers,’ which mediate downstream host responses during virus infection. YTH domain proteins act as readers, recognizing m^6^A-modified RNAs and mediating downstream responses during the interaction of methyltransferases and demethylases [79].

### 3.3. Plant Histones

In plants, genomic DNA is packed into chromatin, with the nucleosome as its functional unit. Each nucleosome contains a core histone octamer, including two copies of four core histone proteins H2A, H2B, H3, and H4 and wrapped by around 147 bp DNA. Histone N-terminal tails undergo covalent modifications, such as methylation, acetylation, ubiquitination, phosphorylation, without altering the underlying DNA sequence [82,83]. Key histone modifications involved in plant epigenetic gene silencing include H3K9, H3K2 and H3K4 methylation [84]. H3K4me3 is an activation mark that can function as a molecular memory, priming plant genes for expression (Figure 4).

H3K9me2 and H3K27me3 act as repressive marks, silencing target genes [85]. Figure 4 illustrates these key histone modifications and their functional interplay in epigenetic gene silencing, including the reinforcing loop between H3K9me2 and DNA methylation, and PRC2-mediated deposition of H3K27me3 for long-term transcriptional repression. H3K9me2 cooperates with DNA methylation to silence viral genes. H3K9 methylation recruits CMT3 to maintain CHG methylation, while DNA methylation in turn recruits SUVH enzymes to sustain H3K9me2, establishing a reinforcing loop that stabilizes epigenetic silencing [86]. Upon viral invasion, the Polycomb Repressive Complex 2 (PRC2) deposits H3K27me3 at specific loci, compacting chromatin and repressing transcription to promote long-term gene silencing. This maintenance is largely independent of DNA methylation or RNA interference pathways, highlighting a distinct regulatory mechanism during viral stress [87]. Recent studies further demonstrate that histone demethylases actively participate in antiviral defense against RNA viruses, dynamically modulating these histone marks to regulate defense-related gene expression and facilitate virus-triggered induced resistance [84].

## 4. Epigenetic and Epitranscriptomic Arms Race Between Plants and Viruses

Plant DNA viruses, such as geminiviruses, are small circular single-stranded DNA (ssDNA) viruses that, upon nuclear entry, are converted into double-stranded DNA and assembled into minichromosomes with host histones, which serve as templates for replication and transcription (Figure 5) [88,89,90,91,92,93]. Viral infection often produces characteristic symptoms, including leaf curling, stunted growth, and mosaic patterns, reflecting the virus’s intimate interaction with host chromatin and cellular machinery [93,94]. In contrast, plant RNA viruses, such as Turnip mosaic virus (TuMV), Cucumber mosaic virus (CMV), Tobacco mosaic virus (TMV), and Alfalfa mosaic virus (AMV), replicate predominantly in the cytoplasm within virus-induced membrane-bound replication complexes and are regulated through epitranscriptomic modifications, including N6-methyladenosine (m^6^A) and small interfering RNA (siRNA)-mediated silencing [95,96]. These differences underscore a nucleus-centric strategy for DNA viruses versus a cytoplasm-centric strategy for RNA viruses, illustrating how virus-host interactions determine replication efficiency, symptom development, and host responses. Both pathways ultimately converge on Argonaute-mediated silencing and small RNA amplification, forming an integrated antiviral network and revealing shared host targets despite distinct subcellular replication niches.

Many DNA viruses encode suppressors that inhibit AGO4-dependent RdDM, reduce siRNA accumulation, or recruit host demethylases such as ROS1 and DME to erase 5-methylcytosine marks on viral minichromosomes [6,97,98]. Similarly, several RNA viruses deploy viral suppressors of RNA silencing (VSRs) that directly bind or inhibit AGO1, which central to PTGS, or sequester siRNAs to block RISC assembly [99,100]. RNA viruses also manipulate the epitranscriptome by encoding AlkB-like RNA demethylases that remove m^6^A marks from viral RNAs, thereby counteracting host-mediated antiviral methylation [27,79,100]. Both DNA and RNA viruses additionally trigger widespread reprogramming of host methylomes and small-RNA regulatory networks, altering CG/CHG/CHH methylation or miRNA pathway components to enhance susceptibility [50,101,102,103,104]. Together, these features reveal how DNA and RNA viruses, despite their distinct replication environments, converge on shared host targets, including AGO proteins, small RNA pathways, and methylation machinery.

Plants employ multifaceted epigenetic and epitranscriptomic mechanisms to restrict viral replication. For DNA viruses such as geminiviruses, transcriptional gene silencing (TGS) mediated by RNA-directed DNA methylation (RdDM) deposits cytosine methylation in CG, CHG, and CHH contexts, effectively repressing viral transcription [105]. Host factors including ARGONAUTE4 (AGO4), DECREASE IN DNA METHYLATION 1 (DDM1), and Ty-1 are critical for this defense, and mutants lacking these components, such as in *Arabidopsis thaliana*, exhibit heightened susceptibility to Cabbage leaf curl virus (CaLCuV) and Beet curly top virus (BCTV) [6,106]. Even low levels of cytosine methylation, naturally or via Ty-1-mediated enhancement, can slow viral replication, facilitate symptom recovery, and amplify antiviral siRNAs. Similarly, in soybean, Mungbean yellow mosaic India virus (MYMIV) DNA shows high intergenic methylation with 24-nt siRNAs indicative of active RdDM [107], while Pepper golden mosaic virus (PepGMV) induces siRNAs that silence viral transcripts and promote recovery in pepper [50,108]. These nuclear DNA virus defenses also feed into cytoplasmic PTGS pathways via small RNA movement, highlighting the functional link and convergence between nuclear TGS and cytoplasmic antiviral silencing.

In parallel, RNA viruses are modulated through epitranscriptomic mechanisms, particularly m^6^A modifications and siRNA-mediated post-transcriptional silencing, which influence RNA stability, translation efficiency, and degradation [95,109]. Figure 6 illustrates plant epigenetic defenses against RNA viruses and viral counter-strategies. m^6^A methylation by writer proteins promotes RNA degradation and suppresses viral replication, whereas erasers remove these marks to facilitate viral RNA translation and infection. Concurrently, the RNA interference pathway generates siRNAs from viral dsRNA, which guide RISC to degrade viral RNAs, highlighting the multilayered host–virus epitranscriptomic arms race. The effect of m^6^A on viral RNAs can be context-dependent: for example, TMV infection reduces m^6^A levels in *Nicotiana tabacum*, whereas AMV infection increases m^6^A in *Arabidopsis thaliana*. In Arabidopsis, the host RNA demethylase ALKBH9B normally removes m^6^A from viral RNAs. Inhibition of ALKBH9B increases m^6^A levels on AMV RNA, which enhances siRNA-mediated antiviral responses and limits systemic virus spread [27]. In contrast, some plant RNA viruses encode their own AlkB-like RNA demethylases that remove m^6^A from viral RNAs, allowing the virus to evade host RNA silencing and promote infection [79]. Similarly, Pepino Mosaic Virus (PepMV) infection in *N. benthamiana* and *S. lycopersicum* modulates m^6^A on viral RNA, altering replication and movement, highlighting the interplay between RNA methylation and viral fitness [79]. RNA virus-induced small RNAs can indirectly influence nuclear DNA methylation or RdDM components, further demonstrating a bidirectional interplay and convergence between cytoplasmic and nuclear antiviral defenses.

Viruses, in turn, have evolved diverse strategies to circumvent host defenses. Geminiviruses actively promote DNA demethylation to evade RdDM-mediated TGS, removing 5-methylcytosine from CG, CHG, and CHH contexts. For instance, the βC1 protein of Tomato yellow leaf curl China betasatellite (TYLCCNB) interacts with ROS1-like DNA glycosylase in *Nicotiana benthamiana* and DEMETER (DME) in *A. thaliana* to reverse viral DNA methylation, while ROS1 also suppresses RdDM, further weakening antiviral defense. Likewise, the C2 protein of Beet severe curly top virus (BSCTV) reduces methylation in promoter repeats of ACCELERATED CELL DEATH 6 (ACD6), increasing host gene expression and reducing resistance [38,97,98]. Figure 5 illustrates how geminiviruses evade plant epigenetic defenses. Viral proteins block RdDM-mediated DNA methylation and siRNA amplification, while host DNA demethylases further enhance viral replication, highlighting the multilayered host–virus arms race. Other viral suppressors, including AC2/AL2 from Tomato golden mosaic virus (TGMV), V2 from TYLCV, Rep, βC1, and C4, disrupt host methylation machinery, RNA silencing, or hormone signaling, facilitating viral replication and symptom development [40,53,98,110]. Field observations, such as partial suppression of RdDM in papaya infected by Tomato leaf curl Joydebpur virus (ToLCJoV), Tomato Leaf Curl New Delhi Virus (ToLCNDV), and Tomato Leaf Curl Bangladesh Virus (ToLCBV) in Bangladesh, or the low Banana Bunchy Top Virus (BBTV) titers in a host-shift to Heliconia in Hawaii, demonstrate the real-world impact of these viral countermeasures [98,102,103,111,112,113]. RNA viruses similarly employ viral suppressors of RNA silencing (VSRs), such as CMV 2b and TRV 16K, which sequester host small RNAs and disrupt RNA-induced silencing complexes (RISC), thereby repressing antiviral gene expression. Additionally, Bean common mosaic virus (BCMV) manipulates host microRNA pathways to alter development and immunity, while the non-coding Hop stunt viroid (HSVd) reduces repressive histone marks such as H3K9me2 to reactivate silenced genes [99,101,114,115]. The durable resistance observed in PRSV-resistant transgenic papaya highlights the potential of exploiting siRNA-mediated epigenetic mechanisms for stable and heritable crop protection [41]. Collectively, these examples highlight both the distinct and overlapping strategies of DNA and RNA viruses, emphasizing a convergent focus on AGO-mediated silencing despite divergent replication compartments. To provide a clear comparative overview of how plant DNA and RNA viruses evade host defenses, we summarized their genome organization, host silencing pathways, viral countermeasures, representative examples, and functional impacts in Table 3. DNA viruses primarily target nuclear methylation machinery, while RNA viruses disrupt cytoplasmic siRNA pathways; both strategies ultimately compromise AGO-mediated host defense.

Plants counter these viral strategies through multilayered strategies, including RNA silencing amplification and targeted phosphorylation of viral proteins. The Ty-1 resistance gene in tomato encodes an RNA-dependent RNA polymerase that enhances cytosine methylation on viral DNA and amplifies viral-derived siRNAs (vsiRNAs), particularly 21- and 22-nt species, thereby promoting degradation of viral RNA even in the presence of suppressors such as C2/TrAP and V2 [106,121] (Figure 5). Complementarily, the Sucrose non-fermenting 1–related kinase 1 (SnRK1) phosphorylates viral proteins to impair their function: βC1 at serine-33 and threonine-78, Rep of TGMV at serine-97, and TrAP of CaLCuV at serine-109. These phosphorylation events collectively reduce viral DNA replication, suppress viral protein activity, and attenuate symptom severity [122,123,124].

Against RNA viruses, plants employ siRNA-mediated post-transcriptional gene silencing. Long viral double-stranded RNAs (dsRNAs) are processed by Dicer-like (DCL) enzymes into 21–25 nucleotide siRNAs, which are unwound into guide and passenger strands. The guide strand is loaded into RNA-induced silencing complexes (RISC) containing Argonaute (AGO) proteins, while the passenger strand is degraded. Activated RISC then targets complementary viral RNAs for cleavage or translational repression, effectively limiting viral replication [24,125]. Examples include CMV in *A. thaliana*, Tobacco rattle virus (TRV), and Tomato mosaic virus (ToMV), highlighting how siRNA amplification and RISC activity suppress viral gene expression. Thus, plants employ complementary nuclear and cytoplasmic defense strategies that converge on small RNA amplification and ARGONAUTE-mediated silencing to combat diverse viral threats.

In summary, plant defense against viruses operates through a multifaceted network of epigenetic and epitranscriptomic mechanisms. Against DNA viruses, hosts employ RdDM-mediated transcriptional gene silencing, maintain or restore viral DNA methylation, amplify antiviral siRNAs, and phosphorylate viral proteins. In turn, geminiviruses deploy silencing suppressors and viral-encoded demethylases to counter these defenses, creating a dynamic molecular arms race that ultimately shapes disease outcomes. RNA viruses, in parallel, are regulated through m^6^A modifications and siRNA-mediated post-transcriptional silencing, with viral countermeasures modulating these pathways to enhance replication and movement. Together, these findings highlight the intricate interplay between host defenses and viral counterstrategies, suggesting that targeted manipulation of epigenetic and epitranscriptomic pathways could provide innovative approaches to enhance crop resistance against both DNA and RNA viruses (Figure 5 and Figure 6). Despite mechanistic differences, both DNA and RNA viruses encounter overlapping host defenses centered on small RNAs and AGO activity, revealing an integrated epigenetic and epitranscriptomic arms race (Figure 5 and Figure 6).

## 5. Future Perspectives and Potential Applications of Epigenetic/Epitranscriptomic Silencing in Plant Virus Management

Epigenetic/epitranscriptomic silencing is emerging as a versatile tool for plant-virus management, offering opportunities in diagnostics, functional genomics, crop improvement, and sustainable disease control. Advances in high-throughput sequencing, epigenomic profiling, and genome editing have enabled precise insights into RNA- and DNA-based epigenetic mechanisms, including small RNA-mediated silencing, DNA and RNA methylation, and chromatin modifications, which collectively shape plant-virus interactions.

### 5.1. Virus Detection and Diagnostic Applications

Virus-derived small interfering RNAs (vsiRNAs) serve as reliable biomarkers for viral infection. High-throughput sequencing of vsiRNAs, known as virus-derived small RNA profiling (vdSAR), enables detection of both known and novel viruses without prior genomic information [126]. DNA virus infections predominantly generate 24-nt siRNAs via RNA-directed DNA methylation (RdDM), whereas RNA viruses produce mainly 21–22 nt vsiRNAs, reflecting post-transcriptional silencing [6,32]. Integration of small RNA profiling with DNA methylation analysis enhances diagnostic sensitivity, though distinguishing viral from host small RNAs and detecting low-abundance signals remain challenges [32,126,127]. Emerging omics technologies, including nanopore-based methylome sequencing, single-cell methylation profiling, and long-read RNA-seq enable single-base resolution of viral DNA/RNA modifications, full-length viral transcript detection, and cell-type-specific host-virus interaction profiling, providing unprecedented sensitivity and specificity for early virus detection [103].

### 5.2. Functional Genomics Using Virus-Induced Gene Silencing

Epigenetic gene silencing, especially via virus-induced gene silencing (VIGS), continues to serve as a useful functional genomics approach in plants [128]. Plant RNA viruses such as PVX (Potato Virus X), TRV (Tobacco Rattle Virus), and CMV (Cucumber Mosaic Virus) are commonly used as vectors in functional genomics. These viruses can induce transcriptional or post-transcriptional gene silencing of reporter transgenes, including GFP and GUS, facilitating functional studies directly in living plants [128]. Recent advancements have optimized VIGS protocols across diverse species, for instance, VIGS has been successfully applied in sunflower using TRV, and in tomato to study ripening-related genes via silencing of a specific methyltransferase [128,129]. A closely relevant recent study demonstrated that short RNA inserts (24–32 nt) delivered through viral vectors can robustly trigger gene silencing [130]. This finding highlights an emerging direction in VIGS optimization, where the use of minimal, precisely designed RNA inserts can significantly increase efficiency and extend VIGS applicability to non-model crops where traditional approaches remain challenging. Recent methylome analyses indicate that viral infections can cause epigenetic modifications, including changes in DNA methylation, histone modifications, and chromatin structure, which are associated with the regulation of defense- and stress-related genes [83,131]. Sequencing technologies, such as enzymatic methyl-seq, allow single-base resolution mapping of these changes, providing information that may help guide research toward virus-resistant cultivars [79].

### 5.3. Engineering Viral Resistance Through Epigenetic Modifications

Epigenetic modifications offer a promising route for engineering durable viral resistance [132]. Some DNA methylation sites and histone changes help plants turn on antiviral genes [133], but viral proteins like AL2 and L2 from Beet Curly Top Virus (BCTV) can block these defenses [54]. By studying these processes, scientists can use epigenetic tools to boost plant immunity and develop viral-resistant crops. Emerging tools such as long-read sequencing and CRISPR/Cas9-based epigenome editing enable precise analysis and targeted manipulation of host epigenetic marks, providing opportunities to enhance plant immunity and generate heritable viral resistance [103,134].

### 5.4. Exogenous Epigenetic-Based Virucides

The exogenous application of double-stranded RNA (dsRNA) or small interfering RNAs (siRNAs) offers a novel, environmentally friendly strategy to control viral infections (Figure 7).

Figure 7 illustrates how applied dsRNA and siRNA trigger antiviral gene silencing in plants. dsRNA is processed by Dicer-like enzymes and loaded into RISC to activate RdDM-mediated transcriptional silencing, while siRNA bypasses Dicer and directly induces epigenetic silencing, effectively preventing viral symptoms. Topical dsRNA treatments targeting viral genomes reduce viral accumulation and symptoms in crops infected with Cucumber Green Mottle Mosaic Virus (CGMMV) and Potato Virus X/Y [135,136]. Recent studies demonstrate that high-pressure dsRNA spraying can induce transcriptional gene silencing via RdDM, a phenomenon termed spray-induced epigenetic modification [119,137]. For example, 24-nt siRNA applications under high pressure induced methylation of the Cauliflower Mosaic Virus (CaMV) 35S promoter in *Nicotiana benthamiana,* leading to transcriptional silencing of viral genes [119,137]. These strategies provide a foundation for sequence-specific antiviral treatments and virucide development.

### 5.5. Ecological and Ethical Considerations

While epigenetic tools offer precise control over viral susceptibility, their ecological and ethical impacts must be considered. Deploying epigenetically modified or RNA-based antiviral crops could alter host-virus dynamics, influence non-target organisms, and affect agroecosystem stability. Regulatory frameworks, risk assessments, and public engagement are essential to ensure responsible application. Integration of biosafety data from field trials of dsRNA sprays and epigenome-edited crops can guide practical deployment, ensuring both effectiveness and ecological compatibility.

Overall, epigenetic gene silencing offers powerful and flexible tools for managing plant viruses. From improving early diagnostics and exploring gene functions to creating durable resistance and environmentally friendly RNA-based treatments, these strategies provide precise and sustainable ways to boost plant immunity. By combining small RNA profiling, methylome mapping, long-read sequencing, and targeted epigenome editing, researchers can deepen our understanding of plant-virus interactions and develop crops with reliable, heritable resistance to viral diseases.

## Figures and Tables

**Figure 1 viruses-18-00017-f001:**
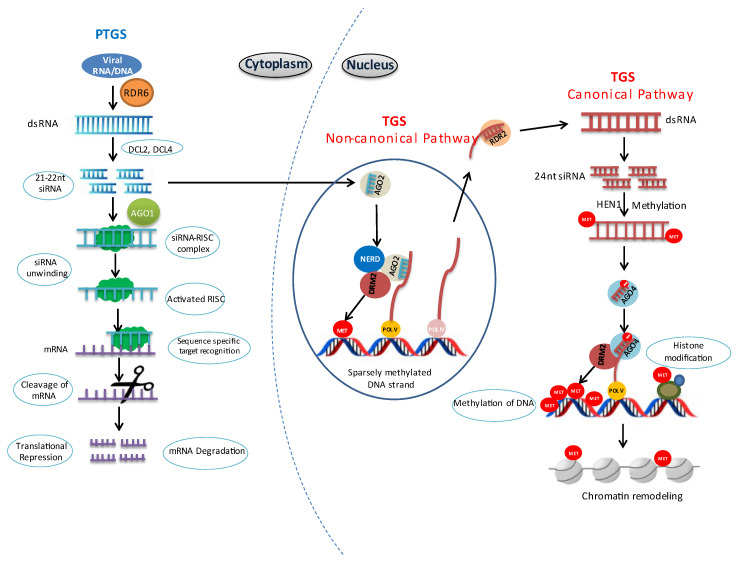
A schematic diagram of post-transcriptional gene silencing (PTGS) and transcriptional gene silencing (TGS) as interconnected antiviral defense pathways in plants. In the left panel, PTGS is illustrated, where viral RNA molecules are recognized and degraded by 21–22 nt small interfering RNAs (siRNAs). These 21–22 nt siRNAs can also feed into a non-canonical RdDM pathway, resulting in sparse and locus-specific DNA methylation. This methylation recruits RNA polymerase IV (Pol IV), which initiates the canonical RdDM pathway. Pol IV transcribes single-stranded RNA (ssRNA), which is then converted into double-stranded RNA (dsRNA) by RNA-DEPENDENT RNA POLYMERASE 2 (RDR2). The dsRNA is processed into 24-nt siRNAs, which guide de novo DNA methylation through ARGONAUTE proteins and associated factors. In addition, histone modifications and chromatin remodeling occur at these loci, further reinforcing transcriptional repression (right panel). Together, these epigenetic changes establish stable TGS and effectively inhibit viral DNA transcription. Arrows indicate the direction of molecular flow; the blue dotted line separates cytoplasmic and nuclear processes.

**Figure 2 viruses-18-00017-f002:**
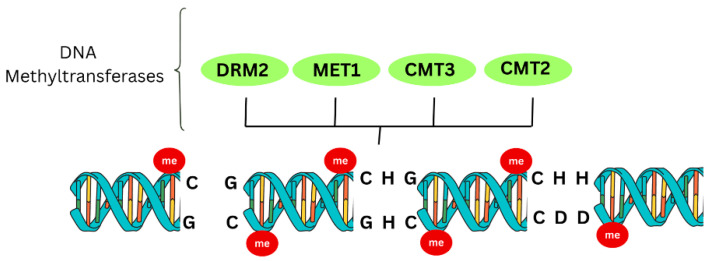
DNA methylation (red circles) occurring in symmetric CG and CHG contexts, as well as asymmetric CHH contexts (where H = A, T, or C). DNA methyltransferases are recruited to specific sequences: MET1 maintains CG methylation, CMT3 catalyzes CHG methylation, while CMT2 and DRM2 establish CHH methylation de novo. The red circles in the figure represent methylated cytosines at these target sites.

**Figure 3 viruses-18-00017-f003:**
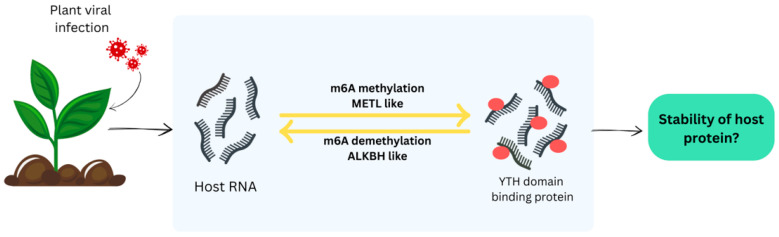
The dynamic regulation of plant viral RNA by m^6^A modification. Viral RNA entering the host is methylated by m^6^A methyltransferases (‘writers’), adding m^6^A marks. These marks can be removed by m^6^A demethylases such as ALKBH9B (‘erasers’), reversing methylation. YTH domain proteins act as ‘readers,’ recognizing m^6^A-modified RNA and mediating downstream responses during host-virus interactions.

**Figure 4 viruses-18-00017-f004:**
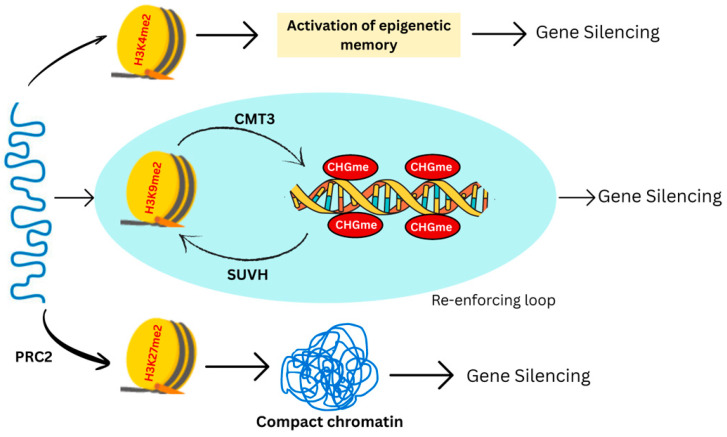
Histone-mediated epigenetic mechanisms in plants. H3K4me2 marks contribute to epigenetic memory and can prime genes for transcriptional regulation. H3K9me2 collaborates with DNA methylation in a reinforcing loop: H3K9me2 recruits CMT3 to maintain CHG methylation, while DNA methylation recruits SUVH enzymes to sustain H3K9me2, collectively reinforcing gene silencing. Additionally, the Polycomb Repressive Complex 2 (PRC2) deposits H3K27me3 at specific loci, compacting chromatin and promoting long-term transcriptional repression.

**Figure 5 viruses-18-00017-f005:**
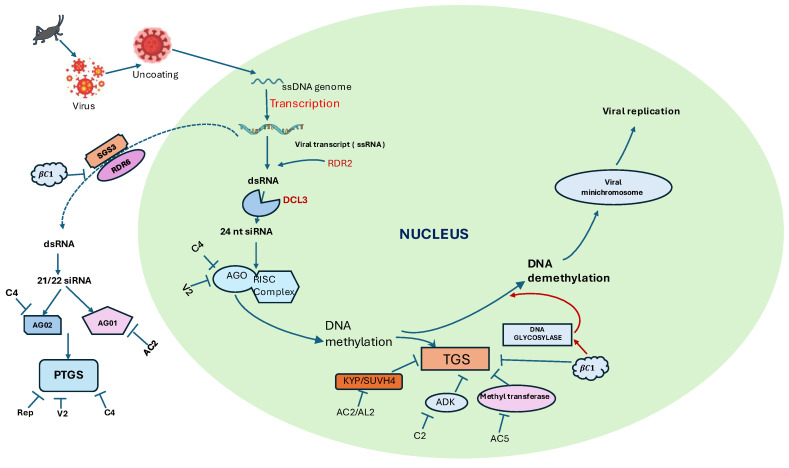
Schematic diagram of geminivirus interactions with plant epigenetic defenses and viral evasion strategies. After entering the plant cell, viral single-stranded DNA (ssDNA) is targeted by post-transcriptional gene silencing (PTGS) in the cytoplasm and transcriptional gene silencing (TGS) in the nucleus. In the nucleus, viral proteins V2 and βC1 interfere with AGO4, blocking RdDM-guided DNA methylation and promoting viral accumulation. Additionally, suppressors, including AC2/AL2, AC4/C4, Rep, and AC5, disrupt TGS by inhibiting key components of the methylation machinery. In the cytoplasm, βC1 degrades SGS3, impairing RDR6-dependent amplification of secondary siRNAs, while C4 also contributes to reduced siRNA accumulation. Furthermore, active DNA demethylation mediated by recruitment of host DNA glycosylases reverses methylation marks on viral genomes, enhancing replication and virulence.

**Figure 6 viruses-18-00017-f006:**
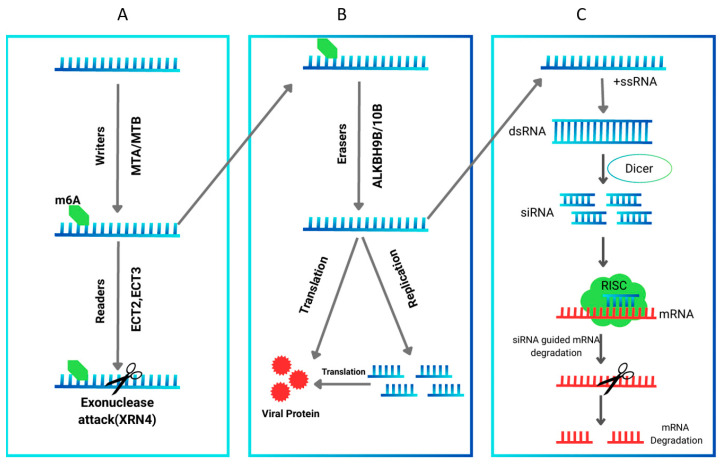
Schematic representation of plant epigenetic defense against RNA viruses and viral counter strategies. (**A**) m^6^A writers catalyze the addition of a methyl group to adenosine residues on viral or host mRNAs, creating an m^6^A mark. m^6^A readers recognize this modification, leading to targeted RNA degradation and suppression of viral replication. (**B**) m^6^A erasers remove methyl groups from mRNAs, allowing translation and replication of viral RNAs, production of viral proteins, and successful infection of plant cells. (**C**) RNA interference pathway: single-stranded viral RNA (ssRNA) is converted into double-stranded RNA (dsRNA), which is processed by Dicer-like (DCL) enzymes into small interfering RNAs (siRNAs). siRNAs are loaded into the RNA-induced silencing complex (RISC), where the guide strand is selected to recognize complementary viral mRNAs, resulting in cleavage, degradation, and epigenetic gene silencing.

**Figure 7 viruses-18-00017-f007:**
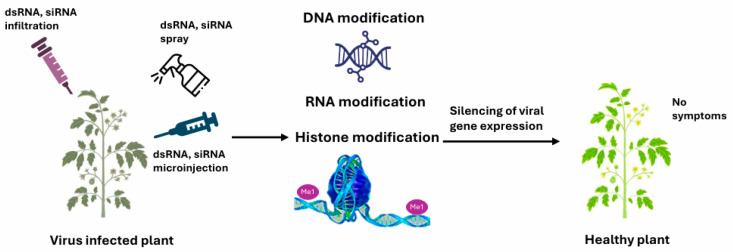
dsRNA and/or siRNA can be applied on virus-infected plants by different approaches such as infiltration, spray, and microinjection. dsRNA is processed by DCL and then associates with RISC, triggering transcriptional gene silencing via the RdDM pathway, so the plant shows no symptoms of virus infection. In contrast, siRNA bypasses DCL, directly binds to RISC, and causes epigenetic gene silencing.

**Table 3 viruses-18-00017-t003:** Comparative synthesis of epigenetic and epitranscriptomic evasion strategies in plant DNA and RNA viruses.

Criteria	DNA Viruses	RNA Viruses	References
Genome type & replication site	dsDNA or ssDNA forming minichromosomes in the nucleus (e.g., Geminiviridae, Caulimoviridae)	ssRNA (+ or − sense) or dsRNA replicating mainly in the cytoplasm (e.g., Potyviridae, Tobamoviridae, Bromoviridae)	[51,116]
Plant silencing mechanism	Transcriptional gene silencing (TGS) via RNA-directed DNA methylation (RdDM) and repressive histone marks on viral minichromosomes	Post-transcriptional gene silencing (PTGS) mediated by siRNAs and RISC targeting viral RNAs; RdDM-mediated TGS generally less relevant	[116,117]
Host factors involved	Pol IV/V, histone methyltransferases (KYP/SUVH), DOMAINS REARRANGED METHYLTRANSFERASES (DRM), RDRs, DCLs producing 24-nt siRNAs, MET1	DCL2/4, RDR6, ARGONAUTE proteins (AGO1/AGO2), RISC complexes, ALKBH9B (RNA demethylase)	[51,116,117,118]
Viral countermeasures	Viral factors (βC1, C2; AC2/AL2, V2, C4) interfere with cytosine methylation, suppress RdDM/TGS, and modulate host histone marks.	Viral factors (VSRs: CMV 2b, TRV 16K; AlkB-like RNA demethylases) modulate host microRNA pathways and alter m^6^A on viral RNA to evade PTGS.	[79,95]
Cross-talk/multi-layered responses	DNA viruses produce RNAs entering PTGS pathways; RNA virus infections can indirectly affect host DNA methylation or RdDM components, triggering multi-layered responses	Experimental use of 24-nt siRNAs or high-pressure dsRNA delivery can induce promoter methylation and TGS, allowing targeting of RNA virus regions under specific conditions	[119,120]
Representative examples	TYLCV, CaLCuV, BCTV, PepGMV	TMV, CMV, AMV, PepMV	[88,95]
Functional impact on virus/host	Slows viral replication, enables partial symptom recovery; countermeasure activity enhances viral transcription	Protects viral RNA from degradation, alters translation efficiency, systemic movement; countermeasures allow evasion of PTGS	[27,91]

## Data Availability

No new data were created or analyzed in this study. Data sharing is not applicable to this article.

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
