# Peer review of "Epigenetic and Epitranscriptomic Antiviral Responses in Plants for Disease Management"

_viruses, 2025, doi:10.3390/v18010017_

Round 1
Reviewer 1 Report
Comments and Suggestions for Authors
Very nice review - congratulations!
Author Response
Dear Reviewers and Editors,
We thank you for your valuable comments on our manuscript. In this revision, we have addressed all the comments, and the changes to the manuscript are clearly identified, and we provide our response to each point raised. We hope that the revisions will meet your expectations and improve the manuscript. Please see our responses for the reviewers' comments:
Reviewer 2
Major revisions:
Question 1: I recommend clearly differentiating between epigenetic (DNA/chromatin-based) and epitranscriptomic (RNA-based) mechanisms, for instance, in light of Ge et al. (2024, Plant Commun doi:10.1016/j.xplc.2024.101232). The authors group processes such as PTGS, m6A, and RNAi under "epigenetic" mechanisms. However, true epigenetics refers to processes that are nucleus-centric and DNA-dependent. PTGS, RNAi, and RNA modifications (e.g., m6A) are better described as epitranscriptomic mechanisms, which can occur in the cytoplasm and are especially critical during RNA virus infections. The authors may consider a more accurate title, e.g., "Epigenetic and epitranscriptomic antiviral responses in plants for disease management." Figure 1 should also be revised accordingly or removed.
Response:
We thank the reviewer for the helpful clarification on distinguishing epigenetic (DNA/chromatin-based) mechanisms from epitranscriptomic (RNA-based) antiviral pathways. We have revised the manuscript accordingly by updating terminology throughout to clearly separate these two regulatory layers, changing the title to “Epigenetic and Epitranscriptomic Antiviral Responses in Plants for Disease Management,” removing Figure 1 to avoid conceptual overlap, and adding the suggested reference (Ge et al., 2024) along with other relevant citations.
Question 2: While geminiviruses are important DNA viruses and excellent models for studying epigenetic interactions, the overemphasis on this virus family may be revised. For instance, the paragraph on Geminiviridae (lines 239-254) could be condensed into one or two sentences. Table 3 and Figure 6 detail geminivirus counter-defense strategies; while informative, these would be significantly strengthened by including parallel examples from RNA viruses.
Response:
We have condensed the discussion of Geminiviridae into two sentences to avoid overemphasis. Additionally, we added representative examples of RNA virus counter-defense strategies in Table 3 .
Question 3: The manuscript would benefit from incorporating recent advances in the field, for instance, discussing that:
3.1: PTGS and mRNA decay are tightly linked in antiviral responses (Trends Plant Sci 2024, doi:10.1016/j.tplants.2023.10.014)
Response:
The proposed reference is not correct. Therefore, we have cited the following reference to support the statement that PTGS and mRNA decay are tightly linked in antiviral responses:
Krzyszton M, Kufel J, and Zakrzewska-Placzek M (2025). RNA interference and turnover in plants—a complex partnership. Front. Plant Sci. 16:1608888. doi: 10.3389/fpls.2025.160888.
Please check line 464.
3.2: L199, plant RNA viruses may encode AlkB-like RNA demethylases (Mol Plant Pathol 2022, doi:10.1111/mpp.13239 ), providing a direct example of viral interference
Response:
We have added information add reviewer suggested. Please check line 653.
3.3: L219, recent work shows that histone demethylases play a role in antiviral defense against RNA viruses, too (BMC Plant Biol 2024, https://doi.org/10.1186/s12870-024-04866-3).
Response:
We have added information add reviewer suggested. Please check line 701
- 3.4: L403, a recent study demonstrated that short RNA inserts (24-32 nt) delivered via viral vectors can trigger robust gene silencing (Plant Biotechnol J 2025, https://doi.org/10.1111/pbi.70254 ), advancing VIGS scalability for functional genomics of non-model crops.
Response:
We did not find the reviewer-suggested reference. Instead, we identified the following relevant study:
García, A., Aragonés, V., Gioiosa, S., Herraiz, F.J., Ortiz-García, P., Prohens, J., Daròs, J.A., and Pasin, F. (2025). Comparative genomics-driven design of virus-delivered short RNA inserts triggering robust gene silencing. bioRxiv, 2025-06.
We now highlighted this study to demonstrate that short RNA inserts (24–32 nt) delivered via viral vectors can robustly trigger gene silencing, thereby enhancing VIGS scalability and functional genomics applications in non-model crops. We have added this information in 6.2. Functional Genomics Using Virus-induced Gene Silencing Please check line 1165 to 1170.
Minor issues:
- The abstract is informative but perhaps too lengthy.
Response:
We thank the reviewer for this observation. The abstract has been revised based on suggestions reviewer 1 and 2
- Remove region-specific mentions, such as "including in Bangladesh" (line 40), for a broader audience.
Response : Removed
- Figures 3, 4, and 5 are missing.
Response:
We have added Figures 3 to 5.
- Table 1: While the table listing viral diseases and economic losses is interesting, it diverges from the molecular focus of the review. Given the emphasis on mechanisms, this table could be omitted or moved to supplementary material.
Response: The table 1 moved to supplementary material.
- The manuscript exhibits inconsistent font sizes and bolding; please revise.
Response: Corrected
- Abbreviations should be revised, e.g., Tobacco rattle virus (TRV) in lines 365 and 406.
Response: Revised
Review 3
MAJOR COMMENTS
- Novelty and Focus
The review summarises familiar concepts effectively but contributes little new analysis. It mainly restates existing reviews (e.g., Deleris et al. 2016; Matzke & Mosher 2014; Pooggin 2013) without introducing a fresh conceptual framework or perspective. To improve its originality, the authors could:
- Include a comparative synthesis table contrasting DNA and RNA virus epigenetic evasion strategies.
Response:
We have included a comparative synthesis table contrasting DNA and RNA virus epigenetic evasion strategies (Table 4).
- Discuss recent omics tools (Nanopore methylome, single-cell methylation, long-read RNA-seq) and how they reshape understanding of viral epigenetic regulation (Hamim et al., 2022 is mentioned but underdeveloped).
Response:
We have incorporated a discussion of recent omics tools, including Nanopore methylome sequencing, single-cell methylation profiling, and long-read RNA sequencing, and how these approaches are reshaping our understanding of viral epigenetic and epitranscriptomic regulation (Hamim et al., 2022).Please check line 1148 to 1154.
- Expand the “Future Perspectives” section into a structured outlook covering diagnostics, biotechnological applications, ecological implications, and ethical considerations.
Response:
We have updated section 6 considering reviewer suggestions. Please check section 6. Future Perspectives and Potential Applications of Epigenetic Gene Silencing in Plant Virus Management.
- Scientific Accuracy and Depth
2.1: Mechanistic descriptions are generally accurate but could benefit from clarifying connections between silencing pathways (PTGS ↔ non-canonical RdDM ↔ TGS).
Response:
We have clarified the connection between silencing pathways (PTGS ↔ non-canonical RdDM ↔ TGS). Please check check line no 445 to 447.
2.2: The RdDM section should clearly differentiate between Pol IV/Pol V pathways and specify AGO family members (AGO4, AGO6, AGO9) to prevent oversimplification.
Response:
We have revised the manuscript and clearly distinguish the canonical RdDM steps mediated by Pol IV (24-nt siRNA biogenesis) and those mediated by Pol V (scaffold transcript production and AGO–siRNA recruitment. We have also specified the major AGO proteins (AGO4, AGO6, and AGO9) involved in guiding DNA methylation within this pathway. The revisions can be found in Section 2.2 (Transcriptional Gene Silencing). Please check line 550 to 553 and line 567 to 570.
Some statements are unsupported or misleading:
2.3: “Epigenetic PTGS viruses that initiate the process are eliminated but silencing in the host is maintained” — this phrase needs supporting primary literature (Jones et al., 1999 is outdated).
Response:
Revised. Please check 455 to 457
2.4: The assertion that DNA methylation patterns can confer resistance to future generations (line 22) is oversimplified; clarify that the stability of methylation across generations is only partial and depends on the context.
Response:
Revised. Please check 20 to 23.
2.5: In the m6A section, include the plant-specific writer complex (MTA–MTB–FIP37–VIRILIZER) and update the reference list with recent work (Yue et al., 2023; Zaccara et al., 2024).
Response
: Revised. Please check 633 to 635
- Integration of RNA and DNA Virus Sections
Sections 4 and 5 read more like parallel mini-reviews than an integrated analysis.
It is suggested to merge them with a comparative sub-section
Response:
We thank the reviewer for this suggestion. In the revised manuscript, Sections 4 and 5 have been combined into a single integrated section titled “Epigenetic and Epitranscriptomic Arms Race Between Plants and Viruses.” We now present a comparative analysis of DNA and RNA viruses, highlighting differences in nucleus- versus cytoplasm-centric defenses while emphasizing shared mechanisms such as ARGONAUTE-mediated silencing and small RNA amplification. Table 4 further summarizes viral genomes, host defense pathways, and viral countermeasures, providing a clear, cohesive overview of the molecular arms race.
3.1 “Convergent and Divergent Epigenetic Strategies between DNA and RNA Plant Viruses.”
Response:
We have revised the manuscript to highlight both the shared and distinct epigenetic strategies of DNA and RNA plant viruses. DNA viruses mainly target nuclear DNA methylation and TGS, while RNA viruses rely on cytoplasmic m⁶A modifications and PTGS. Despite these differences, both converge on small RNA amplification and AGO-mediated silencing, revealing a unified host defense. These comparisons are now clearly presented in the revised Section 4 and summarized in Table 4.
3.2 Please discuss shared suppression mechanisms (e.g., VSRs targeting AGO1/4; demethylases reversing m6A or 5mC marks) and virus-induced reprogramming of host methylomes.
Response:
We have now expanded the section to explicitly compare shared suppression strategies used by RNA and DNA plant viruses..
- Figures and Supplementary Material
Figures (1–8) are informative but lack legends integrated into the main text.
4.1 Each figure should be referenced with a brief explanation of what is novel or schematic.
4.2 Some figures (particularly 2, 6, and 7) closely resemble BioRender templates commonly used in other reviews; check copyright licences and include figure credits in captions.
Reponse:
We have integrated the figure legends into the main text and ensured that each figure is referenced with a brief explanation of its novelty or schematic purpose. All the Figures have been carefully revised, prepared in PowerPoint.
- References
5.1 Extensive but repetitive: several entries are duplicated (e.g., Borges & Martienssen 2015a/b; Cuerda-Gil & Slotkin 2016a/b; Deleris et al. 2016a/b; He et al. 2023a/b). Please tidy the reference list before submission.
Response: We have carefully reviewed the reference list and removed all duplicate entries
5.2 Add recent key papers (2023–2025) on:
Epigenome editing in crops (Li et al., 2024, Plant Biotechnol J.)
m6A regulation in antiviral defense (Xu et al., 2024, Mol Plant-Microbe Interact).
Spray-induced RNA silencing field applications (Cagliari et al., 2024, Front Plant Sci).
Response:
We attempted to locate the specific references suggested (Li et al., 2024, Plant Biotechnol J; Xu et al., 2024, Mol Plant‑Microbe Interact; Cagliari et al., 2024, Front Plant Sci) but were unable to find these exact publications in the stated journals or preprint servers. To address the request, we have instead cited recent, relevant studies from 2023–2025 on epigenome editing in crops, m⁶A regulation in antiviral defense, and spray-induced RNA silencing (Qi et al., 2023; He et al. 2024; Chen et al., 2025), which cover the topics highlighted by the reviewer.
MINOR COMMENTS
Q: Abstract: Replace “plants possess intrinsic molecular defense mechanisms…” with “plants deploy endogenous RNA- and DNA-based epigenetic mechanisms…” for precision.
Response:
We have revised the abstract as suggested. Please check line 14.
Q: Line 46–52: Avoid repetition of “plants adapt to different biotic and abiotic stresses”; merge sentences for flow.
Response:
Revised as suggested. Please check 53 to 67.
Q: 2.1 PTGS: Clarify that RDR6 acts downstream of aberrant RNA recognition; add DCL1 involvement in miRNA biogenesis distinction.
Response:
Revised as suggested. Please check 443 to 451 lines.
Q: 3.2 RNA methylation: Cite examples from host mRNA methylation changes under virus infection (Hu et al., 2022).
Response:
Revised as suggested. Please check 654 to 655 lines.
Q: Section 6: Strengthen link between epigenetic strategies and field-scale applicability (e.g., biosafety of dsRNA sprays).
Response:
Revised. Please check 1386 to 1388 lines
Q: Formatting: Standardise italics for gene/protein symbols (e.g., AGO1, RDR6, DCL3).
Response:
Revised as suggested.
Q: English: Generally clear, but several minor grammatical issues remain (article use, plural consistency, hyphenation of “non-canonical” and “long-term”).
Response:
Revised as suggested.
Reviewer 2 Report
Comments and Suggestions for Authors
The manuscript by Hamim et al. presents a timely and comprehensive review of the molecular mechanisms underlying plant antiviral defense, with a particular focus on epigenetic and epitranscriptomic pathways. The authors highlight key processes such as RNA silencing, DNA methylation, histone modifications, and RNA modifications, linking them to viral resistance and potential biotechnological applications.
While the review covers a broad range of mechanisms and offers valuable insights into potential antiviral strategies, several conceptual and structural concerns need to be addressed to improve it. Specifically:
1. I recommend clearly differentiating between epigenetic (DNA/chromatin-based) and epitranscriptomic (RNA-based) mechanisms, for instance, in light of Ge et al. (2024, Plant Commun doi:10.1016/j.xplc.2024.101232). The authors group processes such as PTGS, m6A, and RNAi under "epigenetic" mechanisms. However, true epigenetics refers to processes that are nucleus-centric and DNA-dependent. PTGS, RNAi, and RNA modifications (e.g., m6A) are better described as epitranscriptomic mechanisms, which can occur in the cytoplasm and are especially critical during RNA virus infections. The authors may consider a more accurate title, e.g., "Epigenetic and epitranscriptomic antiviral responses in plants for disease management." Figure 1 should also be revised accordingly or removed.
2. While geminiviruses are important DNA viruses and excellent models for studying epigenetic interactions, the overemphasis on this virus family may be revised. For instance, the paragraph on Geminiviridae (lines 239-254) could be condensed into one or two sentences. Table 3 and Figure 6 detail geminivirus counter-defense strategies; while informative, these would be significantly strengthened by including parallel examples from RNA viruses.
3. The manuscript would benefit from incorporating recent advances in the field, for instance, discussing that:
- PTGS and mRNA decay are tightly linked in antiviral responses (Trends Plant Sci 2024, doi:10.1016/j.tplants.2023.10.014)
- L199, plant RNA viruses may encode AlkB-like RNA demethylases (Mol Plant Pathol 2022, doi:10.1111/mpp.13239 ), providing a direct example of viral interference with epitranscriptomic regulation.
- L219, recent work shows that histone demethylases play a role in antiviral defense against RNA viruses, too (BMC Plant Biol 2024, https://doi.org/10.1186/s12870-024-04866-3).
- L403, a recent study demonstrated that short RNA inserts (24-32 nt) delivered via viral vectors can trigger robust gene silencing (Plant Biotechnol J 2025, https://doi.org/10.1111/pbi.70254 ), advancing VIGS scalability for functional genomics of non-model crops.
Minor issues:
- The abstract is informative but perhaps too lengthy.
- Remove region-specific mentions, such as "including in Bangladesh" (line 40), for a broader audience.
- Figures 3, 4, and 5 are missing.
- Table 1: While the table listing viral diseases and economic losses is interesting, it diverges from the molecular focus of the review. Given the emphasis on mechanisms, this table could be omitted or moved to supplementary material.
- The manuscript exhibits inconsistent font sizes and bolding; please revise.
- Abbreviations should be revised, e.g., Tobacco rattle virus (TRV) in lines 365 and 406.
Author Response

(The authors gave the same response as above.)

Reviewer 3 Report
Comments and Suggestions for Authors
viruses-3881732
Title: Epigenetic Gene Silencing in Plants: Insights and Applications in Viral Disease Management.
Overall comments:
The manuscript provides a comprehensive review of epigenetic mechanisms (PTGS, TGS, RdDM, m6A, and histone modifications) involved in plant antiviral defence. It integrates classical and recent evidence, highlighting translational applications such as diagnostic tools, VIGS, and dsRNA-based virucides. The topic is timely and pertinent to Viruses, aligning well with the scope of plant–virus interactions. The paper is well organised but occasionally overly detailed and repetitive, with limited critical analysis of mechanisms or current debates in the field. The following major and minor comments aim to enhance the manuscript's readability, originality, and focus.
MAJOR COMMENTS
1. Novelty and Focus
The review summarises familiar concepts effectively but contributes little new analysis. It mainly restates existing reviews (e.g., Deleris et al. 2016; Matzke & Mosher 2014; Pooggin 2013) without introducing a fresh conceptual framework or perspective. To improve its originality, the authors could:
- Include a comparative synthesis table contrasting DNA and RNA virus epigenetic evasion strategies.
- Discuss recent omics tools (Nanopore methylome, single-cell methylation, long-read RNA-seq) and how they reshape understanding of viral epigenetic regulation (Hamim et al., 2022 is mentioned but underdeveloped).
- Expand the “Future Perspectives” section into a structured outlook covering diagnostics, biotechnological applications, ecological implications, and ethical considerations.
2. Scientific Accuracy and Depth
Mechanistic descriptions are generally accurate but could benefit from clarifying connections between silencing pathways (PTGS ↔ non-canonical RdDM ↔ TGS).
The RdDM section should clearly differentiate between Pol IV/Pol V pathways and specify AGO family members (AGO4, AGO6, AGO9) to prevent oversimplification.
Some statements are unsupported or misleading:
“Epigenetic PTGS viruses that initiate the process are eliminated but silencing in the host is maintained” — this phrase needs supporting primary literature (Jones et al., 1999 is outdated).
The assertion that DNA methylation patterns can confer resistance to future generations (line 22) is oversimplified; clarify that the stability of methylation across generations is only partial and depends on the context.
In the m6A section, include the plant-specific writer complex (MTA–MTB–FIP37–VIRILIZER) and update the reference list with recent work (Yue et al., 2023; Zaccara et al., 2024).
3. Integration of RNA and DNA Virus Sections
Sections 4 and 5 read more like parallel mini-reviews than an integrated analysis.
It is suggested to merge them with a comparative sub-section
“Convergent and Divergent Epigenetic Strategies between DNA and RNA Plant Viruses.”
Please discuss shared suppression mechanisms (e.g., VSRs targeting AGO1/4; demethylases reversing m6A or 5mC marks) and virus-induced reprogramming of host methylomes.
4. Figures and Supplementary Material
Figures (1–8) are informative but lack legends integrated into the main text.
Each figure should be referenced with a brief explanation of what is novel or schematic.
Some figures (particularly 2, 6, and 7) closely resemble BioRender templates commonly used in other reviews; check copyright licences and include figure credits in captions.
5. References
Extensive but repetitive: several entries are duplicated (e.g., Borges & Martienssen 2015a/b; Cuerda-Gil & Slotkin 2016a/b; Deleris et al. 2016a/b; He et al. 2023a/b). Please tidy the reference list before submission.
Add recent key papers (2023–2025) on:
Epigenome editing in crops (Li et al., 2024, Plant Biotechnol J.)
m6A regulation in antiviral defense (Xu et al., 2024, Mol Plant-Microbe Interact).
Spray-induced RNA silencing field applications (Cagliari et al., 2024, Front Plant Sci).
MINOR COMMENTS
Abstract: Replace “plants possess intrinsic molecular defense mechanisms…” with “plants deploy endogenous RNA- and DNA-based epigenetic mechanisms…” for precision.
Line 46–52: Avoid repetition of “plants adapt to different biotic and abiotic stresses”; merge sentences for flow.
2.1 PTGS: Clarify that RDR6 acts downstream of aberrant RNA recognition; add DCL1 involvement in miRNA biogenesis distinction.
3.2 RNA methylation: Cite examples from host mRNA methylation changes under virus infection (Hu et al., 2022).
Section 6: Strengthen link between epigenetic strategies and field-scale applicability (e.g., biosafety of dsRNA sprays).
Formatting: Standardise italics for gene/protein symbols (e.g., AGO1, RDR6, DCL3).
English: Generally clear, but several minor grammatical issues remain (article use, plural consistency, hyphenation of “non-canonical” and “long-term”).
Recommendation: Major Revision
The review is solidly informative and relevant, but needs deeper analytical synthesis, updated literature integration, and stylistic polishing before it can stand as a substantial contribution to Viruses.
Author Response

(The authors gave the same response as above.)
